# Experimental Insight into the Structural and Functional Roles of the ‘Black’ and ‘Gray’ Clusters in Recoverin, a Calcium Binding Protein with Four EF-Hand Motifs

**DOI:** 10.3390/molecules24132494

**Published:** 2019-07-08

**Authors:** Sergey E. Permyakov, Alisa S. Vologzhannikova, Ekaterina L. Nemashkalova, Alexei S. Kazakov, Alexander I. Denesyuk, Konstantin Denessiouk, Viktoriia E. Baksheeva, Andrey A. Zamyatnin, Evgeni Yu. Zernii, Vladimir N. Uversky, Eugene A. Permyakov

**Affiliations:** 1Institute for Biological Instrumentation of the Russian Academy of Sciences, Federal Research Center Pushchino Scientific Center for Biological Research of the Russian Academy of Sciences, 142290 Pushchino, Russia; 2Department of Biomedical Engineering, Pushchino State Institute of Natural Sciences, 142290 Pushchino, Russia; 3Structural Bioinformatics Laboratory, Biochemistry, Faculty of Science and Engineering, Åbo Akademi University, 20520 Turku, Finland; 4Pharmaceutical Sciences Laboratory, Pharmacy, Faculty of Science and Engineering, Åbo Akademi University, 20520 Turku, Finland; 5Belozersky Institute of Physico-Chemical Biology, Lomonosov Moscow State University, 119992 Moscow, Russia; 6Institute of Molecular Medicine, Sechenov First Moscow State Medical University, 119991 Moscow, Russia; 7Department of Molecular Medicine and USF Health Byrd Alzheimer’s Research Institute, Morsani College of Medicine, University of South Florida, Tampa, FL 33612, USA

**Keywords:** calcium binding proteins, EF-hand, recoverin, protein structure, protein function

## Abstract

Recently, we have found that calcium binding proteins of the EF-hand superfamily (i.e., a large family of proteins containing helix-loop-helix calcium binding motif or EF-hand) contain two types of conserved clusters called cluster I (‘black’ cluster) and cluster II (‘grey’ cluster), which provide a supporting scaffold for the Ca^2+^ binding loops and contribute to the hydrophobic core of the EF-hand domains. Cluster I is more conservative and mostly incorporates aromatic amino acids, whereas cluster II includes a mix of aromatic, hydrophobic, and polar amino acids of different sizes. Recoverin is EF-hand Ca^2+^-binding protein containing two ‘black’ clusters comprised of F35, F83, Y86 (N-terminal domain) and F106, E169, F172 (C-terminal domain) as well as two ‘gray’ clusters comprised of F70, Q46, F49 (N-terminal domain) and W156, K119, V122 (C-terminal domain). To understand a role of these residues in structure and function of human recoverin, we sequentially substituted them for alanine and studied the resulting mutants by a set of biophysical methods. Under metal-free conditions, the ‘black’ clusters mutants (except for F35A and E169A) were characterized by an increase in the α-helical content, whereas the ‘gray’ cluster mutants (except for K119A) exhibited the opposite behavior. By contrast, in Ca^2+^-loaded mutants the α-helical content was always elevated. In the absence of calcium, the substitutions only slightly affected multimerization of recoverin regardless of their localization (except for K119A). Meanwhile, in the presence of calcium mutations in N-terminal domain of the protein significantly suppressed this process, indicating that surface properties of Ca^2+^-bound recoverin are highly affected by N-terminal cluster residues. The substitutions in C-terminal clusters generally reduced thermal stability of recoverin with F172A (‘black’ cluster) as well as W156A and K119A (‘gray’ cluster) being the most efficacious in this respect. In contrast, the mutations in the N-terminal clusters caused less pronounced differently directed changes in thermal stability of the protein. The substitutions of F172, W156, and K119 in C-terminal domain of recoverin together with substitution of Q46 in its N-terminal domain provoked significant but diverse changes in free energy associated with Ca^2+^ binding to the protein: the mutant K119A demonstrated significantly improved calcium binding, whereas F172A and W156A showed decrease in the calcium affinity and Q46A exhibited no ion coordination in one of the Ca^2+^-binding sites. The most of the N-terminal clusters mutations suppressed membrane binding of recoverin and its inhibitory activity towards rhodopsin kinase (GRK1). Surprisingly, the mutant W156A aberrantly activated rhodopsin phosphorylation regardless of the presence of calcium. Taken together, these data confirm the scaffolding function of several cluster-forming residues and point to their critical role in supporting physiological activity of recoverin.

## 1. Introduction

Calcium binding proteins take part in all biological processes. One of the most important families of calcium binding proteins is the EF-hand protein superfamily (reviewed in [1,2,3]). The EF-hand calcium binding domain consists of two helices and a calcium binding loop between them [4,5,6]. Most of the EF-hand family members, such as calmodulin and recoverin, function as Ca^2+^-dependent sensor proteins. Others, mostly parvalbumin and calbindin, seem to serve as cytosolic Ca^2+^ buffers [1].

The EF-hand domains in Ca^2+^ binding proteins are usually paired. In our previous work, we found two highly conserved structural motifs, which provide a supporting scaffold for the Ca^2+^ binding loops and give a contribution to formation of the EF-hand domain hydrophobic core [7]. Each structural motif forms a cluster of three amino acids. These clusters were called cluster I (‘black’ cluster) and cluster II (‘grey’ cluster). Cluster I (‘black’) is much more conservative and mostly incorporates aromatic amino acids. It lacks destabilizing interactions and has a predominant aromatic mini-core that is stabilized by a set of linked CH-π and CH-O hydrogen bonds. We suggested that cluster I is vital for structural stabilization of the EF-hand domain in its critical gate region, where the polypeptide chain enters and exits the domain. In contrast, cluster II includes a mix of aromatic, hydrophobic, and polar amino acids of different sizes. It lacks stabilizing interactions and more often forms destabilizing interactions. We suggested that the higher variability of cluster II (‘gray’) could promote adaptation of a protein to the conformational and dynamic requirements imposed by the need to ensure wide range of kinetic and equilibrium metal binding constants, as well as recognition of various targets (proteins, lipids and so on). The analysis of the structures of clusters I and II and of their rearrangements in response to Ca^2+^ binding enabled us to propose a more detailed classification of the EF-hand proteins, different from the conventional division into metal sensors and buffers [7]. Currently, the availability of structural and functional information on the importance of the discovered ‘black’ and ‘grey’ clusters in calcium binding proteins is limited. Hence, these clusters should be subjected to the focused experimental analyses. To address this issue, we performed a systematic analysis of the ‘black’ and ‘grey’ clusters in different calcium binding proteins. The present work is devoted to the experimental characterization of the roles of the ‘black’ and ‘grey’ clusters in a four EF-hand motif-containing calcium binding protein, recoverin.

Recently, we have studied the ‘black’ and ‘gray’ clusters in rat β-parvalbumin (109 residue-long calcium binding protein containing two EF-hand domains) comprised of F48, A100, F103 and G61, L64, M87, respectively [8], and in human calcium binding S100P protein (95 residue-long protein containing two EF-hand domains), comprised of residues F15, F71, and F74 and L33, L58, and K30, respectively [9]. Ala scanning of these amino acid residues demonstrated that amino acids of the cluster I in both proteins provide more essential contribution to the maintenance of structural and functional properties of these proteins in comparison with the residues of the cluster II.

Proteins that contain four EF-hand motifs are the most abundant group of calcium-binding proteins within the EF-hand family. These four EF-hand motif-containing proteins include several well-known EF-hand proteins, such as calmodulin, muscle contractile protein troponin C, myosin light chains, and all members of the neuronal Ca^2+^ sensor (NCS) protein family (recoverin, frequenin, and others) [10]. These proteins are made up of two globular domains (the N- and C-terminal domains) connected by a linker of varying length. Each domain contains a pair of the EF-hand motifs. The structures of myristoylated recoverin in solution with 0, 1, and 2 Ca^2+^ bound have been determined by NMR spectroscopy [11]. Recoverin has an overall globular shape with the two domains in close proximity, connected through a short U-shape linker, rather than the long central helix providing the classic dumbbell shape of calmodulin.

Recoverin has a myristoyl group at its N terminus [12], which, in the Ca^2+^-free state, is buried in a hydrophobic region of the protein [13]. Ca^2+^ binding to the two functional EF-hands of recoverin located in its N-terminal (EF-hand 2) and C-terminal (EF-hand 3) domains [14], results in a transfer of the myristoyl group to the protein surface in contact with the solvent (Ca^2+^-myristoyl switch). It is a multistep process, in which the EF-hands are sequentially filled by Ca^2+^ and recoverin undergoes a major conformational change [11,15,16,17,18,19]. Due to the Ca^2+^ binding-induced emergence of the solvent-exposed myristoyl group, recoverin can interact with membranes and the interaction appears to be driven mainly by hydrophobic forces [19,20]. In the Ca^2+^-bound state, recoverin functions as an inhibitor of rhodopsin kinase [19,21,22,23], and this feature is thought to contribute to light adaptation of photoreceptor cells [19,24].

In the present work, we sequentially substituted all amino acid residues of the ‘black’ and ‘gray’ clusters in recombinant wild type human recoverin by alanine and studied physicochemical and functional properties of the resulting mutants.

## 2. Results and Discussion

Human recoverin has four EF-hand calcium-binding motifs and therefore it contains two ‘black’ clusters composed of F35, F83, Y86 (N-terminal domain) and F106, E169, F172 (C-terminal domain) as well as two ‘gray’ clusters composed of F70, Q46, F49 (N-terminal domain) and W156, K119, V122 (C-terminal domain) (Figure 1). To investigate a role of these residues in recoverin in maintaining of its structure and function, we sequentially substituted them by Ala. The resulting mutants were expressed in *Escherichia coli* and their molecular masses were confirmed by means of mass spectrometry. All the mutants obtained were myristoylated; the fraction of non-myristoylated forms in the protein samples did not exceed 16%.

To begin with, we investigated effects of the cluster mutations on the secondary structure of recoverin by means of circular dichroism (CD) spectroscopy. Particularly, we measured far-UV CD spectra of human recoverin and its cluster mutants in apo (1 mM ethylenediaminetetraacetic acid (EDTA)-KOH), Mg^2+^-loaded (1 mM ethylene glycol-bis(β-aminoethyl ether)-*N,N,N′,N*′-tetraacetic acid (EGTA)-KOH, 1 mM MgCl_2_, pH 7.4) and Ca^2+^-loaded (1 mM CaCl_2_) states at 15 °C (Appendix A).

The data on secondary structure content estimated from the CD spectra are summarized in Table 1 and Figure 2. As can be seen, the most of the alanine substitutions in the ‘black’ clusters with the exception of E169A resulted in a slight increase in the α-helical content of apo-proteins (Figure 2A). In contrast, the mutations in the ‘gray’ clusters except for K119A caused a slight decrease in α-helical content in apo-proteins (Figure 2B). The changes did not exceed 2%, which is close to the accuracy of the α-helical content evaluation. Surprisingly, all the Ca^2+^-loaded mutants were characterized by an increase in the α-helical content, which in some cases exceeded 5% (Figure 2). The most pronounced increase in α-helical content was induced by mutations F83A, Y86A (N-terminal ‘black’ cluster) and V122A (C-terminal ‘gray’ cluster). Next, we studied effects of the mutations on multimerization/aggregation of recoverin by a chemical crosslinking method (Appendix A and Table 2). Apo- and Mg^2+^-loaded forms of recombinant wild type (rWT) recoverin and all its mutants contained only monomers (60–70%) and dimers. Meanwhile, the binding of calcium to these proteins resulted in a decrease of the monomer content down to 15–40% and appearance of trimers and higher order multimers.

Figure 3 demonstrates effects of the cluster mutations on the monomer content for various states of recoverin. It is seen that for apo- and Mg^2+^-loaded states of recoverin the alanine substitutions in the ‘black’ and ‘gray’ clusters except for K119A only slightly changed the monomer fraction (by 2–6%). However, in the presence of Ca^2+^, the substitutions in the ‘black’ and ‘gray’ clusters of N-terminal domain of recoverin significantly increased its monomer content (by 16–25% in comparison with the rWT protein) indicating dramatically reduced susceptibility of the protein to multimerization and aggregation. Thus, the surface properties of Ca^2+^-bound recoverin seems to be sensitive to the presence of N-terminal clusters residues.

The far-UV CD spectroscopy was further employed to investigate the thermal stability of apo-forms of rWT recoverin and its cluster mutants. Figure 4 demonstrates temperature dependences of the molar ellipticity at 208 nm for these proteins measured in the absence of metal ions. A decrease in ellipticity associated with a thermally-induced protein unfolding was analyzed in the temperature range from 40 to 70 °C as the subsequent heating was accompanied by noticeable protein aggregation.

Table 3 and Figure 5 summarize the mid-temperatures of the thermal transitions of apo-states of rWT human recoverin and its cluster mutants. One can see that several mutations in the ‘black’ and ‘gray’ clusters of the C-terminal domain of recoverin caused a pronounced decrease in thermal stability of the protein. Thus, alanine substitutions of F172 (C-terminal ‘black’ cluster) as well as W156 and K119 (C-terminal ‘gray’ cluster) resulted in an 8–16 °C shift of the thermal transition to lower temperatures. The W156A mutant (C-terminal ‘gray’ cluster) was characterized by the lowest thermal stability (shift of the thermal transition to lower temperatures by 16 °C).

Generally, the mutations in the N-terminal domain of recoverin caused less pronounced changes in thermal stability compared with the mutations in the C-terminal domain. In addition, the mutations in the C-terminal domain of recoverin commonly caused a decrease in its thermal stability, whereas the substitutions in N-terminal domain of the protein resulted both in a decrease (F83A) and a slight increase (by 2–3 °C) (mutants Y86A, F70A, and F49A) in thermal stability. Among the N-terminal mutants, F83A was recognized as the most susceptible to thermal denaturation as it exhibited a decrease in unfolding transition temperature by 7.5 °C.

In the following step, we determined calcium affinities of rWT recoverin and its cluster mutants at 20 °C by means of spectrofluorometric Ca^2+^-titrations using the appropriate Ca^2+^-buffer solutions (Figure 6). The binding of calcium to rWT recoverin caused an increase in its tryptophan fluorescence intensity at 350 nm and a red shift of fluorescence spectrum maximum by 12 nm (Figure 6 and Figure 7). Generally, such Ca^2+^-induced effect depends on the quenching and relaxation properties of the environment of tryptophan chromophores in apo- and Ca^2+^-loaded forms of a protein (reviewed in [25]). In recoverin, this effect is explained by a transfer of buried tryptophan residues to the protein surface in contact with water molecules. Interestingly, the alanine substitutions in the ‘black’ cluster of the N-terminal domain of recoverin changed the sign of the Ca^2+^-induced effect: the binding of calcium to F35A, F83A, and Y86A mutants decreased the fluorescence intensity at 350/370 nm (Figure 6a). Similar effect was observed for F49A mutant containing substitution in‘gray’ cluster of the N-terminal domain (Figure 6b). In all other cases, the binding of calcium resulted in an increase in the fluorescence intensity at 320/350 nm (Figure 6c–e). The most complex effect was obsreved in the case of alanine substitution of K119 in the C-terminal domain as in this case the essential decrease in the amplitude of the Ca^2+^-induced fluorescence change was accompanied by its shift to significnatly lower calcium concentrations; i.e., by essential increase in calcium affinity.

The experimental data on calcium titrations of the most of recoverin forms were well approximated by theoretical curves computed according to both cooperative and sequential binding schemes (Table 4).

However, for some of the mutants, the data were fitted only to cooperative (F172A and W156A) or sequential (Y86A, F70A) binding models indicating that the introduction of these substitutions not only altered Ca^2+^-binding to individual EF-hands, but also affected interplay between these motifs. For instance, in the case of F70A mutant (substitution in the ‘gray’ cluster of the N-terminal domain) the filling of the first Ca^2+^-binding site caused a decrease in fluorescence intensity, whereas the filling of the second site resulted in an increase in fluorescence intensity (Figure 6e). In addition, the data for the mutant Q46A (‘gray’ cluster of the N-terminal domain) were best fitted only to the one-site binding scheme (Figure 6e), suggesting that this protein exhibited no ion coordination in one of the Ca^2+^-binding sites. Figure 8 and Table 4 show effects of cluster mutations on Ca^2+^-induced change in free energy of recoverin reflecting its affinity to the cation. Like in the case of thermal denaturation, the most pronounced changes were caused by the alanine substitutions of F172 (‘black’ cluster of the C-terminal domain), W156 and K119 (‘gray’ cluster of the C-terminal domain) as well as Q46 (‘gray’ cluster of the N-terminal domain). Yet, the effects of these mutations were differently directed: the mutant K119A demonstrated significantly improved calcium binding (three to four orders of magnitude), whereas F172A and W156A mutants were characterized by the 10-fold decrease in calcium affinity.

In order to evaluate impact of the cluster mutations on functional properties of recoverin, we next studied the interaction of recoverin and its mutants with photoreceptor membranes and examined effects of these proteins on phosphorylation of rhodopsin by GRK1 (rhodopsin kinase). It is known that Ca^2+^ binding to functional EF-hands of recoverin (EF-hand 2 and EF-hand 3 [14]) results in opening of hydrophobic pocket of the protein and a transfer of its initially buried myristoyl group to the solvent (Ca^2+^-myristoyl switch) [27]. The exposed residues of the hydrophobic pocket (F23, W31, F35, F49, I52, Y53, F56, Y86, and L90) form a binding site for N-terminal helix of GRK1 [28], whereas myristoyl group mediate the interaction of the protein with photoreceptor membranes [19,20]. Our experiments revealed that the most of the N-terminal clusters mutations (except for F49A) suppressed membrane binding of Ca^2+^-loaded recoverin (Figure 9A) and its inhibitory activity towards GRK1 (Figure 9B). We suggest that these effects are connected to each other as the reduced membrane binding of recoverin might decrease its effective concentration in the proximity to GRK1/activated rhodopsin. In addition, the negative effect of the substitutions of F35, Y86 (N-terminal ‘black’ cluster), and F49 (N-terminal ‘gray’ cluster) on inhibitory activity of recoverin may be attributed to their participation in the hydrophobic pocket responsible for GRK1 binding [28]. In contrast to N-terminal mutations, the substitutions in C-terminal clusters generally did not affect membrane binding of recoverin (Figure 9A). Consistently, the majority of the mutants with the exception of F172A and W156A exhibit effective GRK1 inhibition (Figure 9B). Surprisingly, the mutant W156A aberrantly activated rhodopsin phosphorylation regardless of the presence of calcium (Figure 9B and Figure 10B). The origin of this effect remains puzzling, although it can be associated with increased membrane binding of both Ca^2+^-free and Ca^2+^-bound forms of the protein and/or promoting formation of a complex between GRK1 and bleached rhodopsin.

The aforementioned multiparametric analysis considered the effects of cluster mutants on structural and functional properties and conformational stability of human recoverin. Since the peculiarity of calcium binding to recoverin, interaction of this protein with rhodopsin kinase, and its global conformational stability are all dependent on the recoverin tertiary structure, the results of these analyses indicated the effects of said mutations on the overall protein fold. Obviously, in addition to the global structural changes causing alterations of the protein functionality and stability, point mutations might affect local predisposition of a polypeptide chain for folding.

Such local alterations can be evaluated by analyzing the effects of mutations on protein intrinsic disorder predisposition. Therefore, to check if the cluster mutations influence the intrinsic disorder predisposition of human recoverin (UniProt ID: P35243), we first applied several commonly used disorder predictors to analyze the global intrinsic disorder status of this protein. Here we utilized members of the PONDR family, such as PONDR^®^ VLXT, PONDR^®^ VL3, PONDR^®^ VSL2, and PONDR^®^ FIT, and two forms of the IUPred algorithm for finding short and long intrinsically disordered regions. Results of this analysis are summarized in Figure 11A, which clearly demonstrates that intrinsic disorder predisposition is non-homogeneously distributed within the sequence of human recoverin. In fact, this protein was predicted to have a disordered N-tail (residues 1–15) followed by an ordered region (residues 16–139) containing three EF-hand motifs (residues 25–60, 61–96, and 97–132), a disordered linker (residues 140–147), moderately ordered region (residues 148–189) containing EF-hand motif #4 (residues 147–182), and ending with a disordered C-tail (residues 190–200). Figure 11A shows that the N-terminal two thirds of this protein is noticeably less disordered than its C-terminal third. Although all four EF-hand domains were located within the more ordered parts of human recoverin, linkers connecting these functional motifs are either disordered or at least highly flexible. Figure 11A also shows that the residues from the both ‘black’ clusters are located within more disordered regions than the residues of the both ‘gray’ clusters.

Figure 11B shows the effect of the cluster mutations (F35A, F83A, and Y86A in the N-terminal ‘black’ cluster, F70A, Q46A, and F49A in the N-terminal ‘gray’ cluster, F106A, E169A, and F172A in the C-terminal ‘black’ cluster, and W156A, K119A, and V122A in the C-terminal ‘gray’ cluster) on the local intrinsic disorder propensity of human recoverin evaluated using the PONDR^®^ VSL2 algorithm. To better visualize the mutation-induced alterations in the local intrinsic disorder propensity, Figure 11C shows “difference disorder spectra” calculated by the subtracting the PONDR^®^ VSL2 disorder profile of the wild type human recoverin from the disorder profiles of each of it cluster mutants. Here, positive and negative values correspond to the mutations that induce increase and decrease in local disorder predisposition, respectively. Figure 11C illustrates that 9 of 12 cluster mutations were predicted to cause increase in the local intrinsic disorder propensity. The exceptions are given by Q46A mutation in the N-terminal ‘gray’ cluster, K119A mutation in the C-terminal ‘gray’ cluster, and E169A mutation in the C-terminal ‘black’ cluster. These results were rather expected since these three mutations (Q46A, K119A, and E169A) resulted in changes of more disorder-promoting residues Q, K, and E to a disorder-neutral alanine, whereas in all other cases, alanine substitutes residues with high order-promoting potential (F, Y, W, and V).

Figure 11B,C also show that typically, mutations in the N-terminal half of the protein generate less perturbations in the intrinsic disorder propensity than mutations in the C-terminal half of recoverin. Furthermore, the mutations in ‘black’ clusters caused a bit more noticeable changes in the local disorder propensity of recoverin in comparison with the effects of the mutations in the corresponding ‘gray’ clusters. These observations are in some agreement with the results of the analogous analyses conducted earlier to evaluate the effects of cluster mutations on intrinsic disorder predispositions of the rat β-parvalbumin [8] and human S100P [9]. However, the differences in the effects of mutations in ‘black’ and ‘gray’ clusters on local intrinsic disorder predisposition of rat β-parvalbumin and human S100P were more pronounced that those found here for the cluster mutations of human recoverin.

Finally, Figure 12 represents the results of the comparison of changes induced by various cluster mutations in the local intrinsic disorder propensity of human recoverin with the mutation-induced changes in helical propensity and thermal stability of this protein. This analysis revealed that these three parameters are weakly correlated, indicating that the intrinsic disorder predisposition of the residues of ‘black’ and ‘gray’ clusters may play some role in controlling the structure and conformational stability of human recoverin.

Recently, we have studied the ‘black’ and ‘gray’ clusters in rat β-parvalbumin (109 residue-long calcium binding protein containing two EF-hand domains) comprised of F48, A100, F103 and G61, L64, M87, respectively [8]. These amino acid residues were sequentially substituted by Ala, except Ala100, which was substituted by Val. Physical properties of the mutants were studied by circular dichroism, scanning calorimetry, dynamic light scattering, chemical crosslinking, and fluorescent probe methods. The Ca^2+^ and Mg^2+^ binding affinities of these mutants were evaluated by intrinsic fluorescence and equilibrium dialysis methods. It was found that the alanine substitutions in the cluster I of rat β-parvalbumin caused noticeably more pronounced changes in various structural parameters of the protein, such as hydrodynamic radius of its apo-form, thermal stability of Ca^2+^/Mg^2+^-loaded forms, and total energy of Ca^2+^ binding in comparison with the changes caused by similar amino acid substitutions in the cluster II. Computational analysis of the effects of these mutations on the intrinsic disorder predisposition of rat β-parvalbumin showed that local intrinsic disorder propensities and the overall levels of predicted disorder are strongly affected by mutations in the cluster I, whereas mutations in cluster II have less pronounced effects. These results demonstrate that amino acids of the cluster I in rat β-parvalbumin provide more essential contribution to the maintenance of structural and functional properties of the protein in comparison with the residues of the cluster II [8].

In human calcium binding S100P protein (95 residue-long protein containing two EF-hand domains), the ‘black’ and ‘gray’ clusters include residues F15, F71, and F74 and L33, L58, and K30, respectively. To evaluate the effects of these clusters on structure and functionality of human S100P, we performed Ala scanning [9]. The resulting mutants were studied by a multiparametric approach that included circular dichroism, scanning calorimetry, dynamic light scattering, chemical crosslinking, and fluorescent probes. It was found that the alanine substitutions in the clusters I and II caused comparable changes in the S100P structural properties. However, analysis of heat- and GuHCl-induced unfolding of these mutants showed that the alanine substitutions in the cluster I caused notably more pronounced decrease in the protein stability compared to the changes caused by alanine substitutions in the cluster II [9].

## 3. Materials and Methods

### 3.1. Materials

Molecular biology grade HEPES, ultra-grade H_3_BO_3_/glycine/Tris&MES were from Calbiochem (San Diego, CA, USA), Fluka (Seelze, Germany), Sigma-Aldrich Co. (St. Louis, MO, USA), and Amresco (Solon, OH, USA), respectively. Pharma grade KCl and ultra-grade TCA were purchased from Panreac AppliChem (Darmstadt, Germany). Biotechnology grade DTT was purchased from DiaM (Moscow, Russia). Biotechnology grade 2-ME and molecular mass markers for SDS-PAGE were from Helicon (Moscow, Russia). Ultra-grade KOH, EDTA and standard solutions of CaCl_2_ were purchased from Sigma-Aldrich Co. Standard solutions of EDTA potassium salt was prepared as described in [29]. Molecular biology grade glutaric aldehyde was from Amersham Biosciences (Little Chalfont, United Kingdom). Biochemistry grade Coomassie Brilliant Blue R-250 was products of Merck (Kenilworth, NJ, USA). SP Sepharose Fast Flow was a product of Amersham Biosciences Co. Toyopearl SuperQ-650M was purchased from Tosoh Bioscience (San Francisco, CA, USA). Sephadex G-25 and PD MidiTrap^TM^ G-25 were products of Pharmacia LKB (Uppsala, Sweden) and GE Healthcare (Chicago, IL, USA), respectively. D-10-camphorsulfonic acid was from JASCO, Inc. (Easton, PA, USA). Other chemicals were reagent grade or higher.

All buffers and other solutions were prepared using ultrapure water (Millipore Simplicity 185 system). Plastic or quartz ware was used instead of glassware, to avoid contamination of protein samples with Ca^2+^. DTT solutions were prepared using degassed buffers immediately prior to the usage to avoid DTT oxidation. Thermo SnakeSkin dialysis tubing (3.5 kDa MWCO, ThermoFisher Scientific (Waltham, MA, USA)) and Millipore Amicon Ultra centrifugal filters (3.0 kDa MWCO, MilliporeSigma (Burlington, VT, USA)) were used for dialysis and concentration of protein solutions.

### 3.2. Expression and Purification of Human Wild-Type Recoverin and Its Mutants

Human recoverin cDNA was produced using total RNA extract from Y79 retinoblastoma cell line and oligonucleotide ACAGCTGAACAGTTGGCA. The obtained cDNA was exploited further as a template for DNA amplification using a pair of oligonucleotides (5’-ATACCATGGGGAACAGCAAAAGTGGGGC and 5’-TATAGTCGACTCAGGCGTTCTTCA TCTTTTC). The resulting product containing human recoverin gene was subcloned into pET-15b(+) expression vector (Novagen, Germany) using NcoI and XhoI. Recombinant wild type (rWT) recoverin was produced in *E. coli* and purified in reducing conditions as previously described [30,31] with minor modifications. Recombinant myristoylated form of recoverin was expressed in the *E. coli* strains pET15b rec/pBB131/BL21(DE3)RILLC+. The cells were grown for 5 h, harvested by centrifugation at 5000× *g* for 15 min at 4 °C, resuspended in 50 mL of lysis buffer (50 mM Tris-HCl, pH 8.0, 100 mM NaCl, 1 mM DTT, 1 mM EDTA) and disintegrated using a French press. The lysate was centrifuged at 14,000×*g* for 30 min at 4 °C. The supernatant was adjusted to 3 mM CaCl_2_, filtrated through cotton wool and applied to a phenyl-Sepharose column (3 × 9 cm) equilibrated with buffer A (20 mM Tris-HCl, pH 8.0, 1 mM DTT, 2 mM CaCl_2_). The column was washed with buffer A (flow rate of 5 mL/min) and recoverin was eluted with buffer A, containing 2 mM EDTA. The recoverin-containing fraction was then loaded on a Toyopearl SuperQ-650M column (0.9 cm × 7.5 cm) equilibrated with 40 mM Tris-HCl, pH 8.0, 1 mM DTT. Recoverin was eluted with a linear gradient of 0–300 mM NaCl. The samples of purified rWT human recoverin samples were exhaustively dialyzed at 4 °C against 20 mM Tris-HCl, pH 8.0, 1 mM DTT, concentrated by Amicon Ultra-15 Centrifugal Filter Unit with Ultracel-3 membrane at 7000×*g* for 20 min at 4 °C, freeze-dried and stored at −70 °C. Purity of the protein samples was confirmed by native and SDS-PAGE. The degree of myristoylation was determined by analytical HPLC on apHera ™ C18 polymeric reversed phase HPLC column: it exceeded 89% for the myristoylated protein, and does not exceeded 11% for non-myristoylated recoverin. Recoverin concentration was measured spectrophotometrically using a molar extinction coefficient at 280 nm of 25,440 M^−1^cm^−1^, as given by Expasy ProtParam tool (https://web.expasy.org/protparam/).

The genetic constructs encoding recoverin mutants with substitutions of F35, F83, Y86, F106, E169, F172, F70, Q46, F49, W156, K119 and V122 by Ala were produced by site-directed mutagenesis [32] using synthetic oligonucleotides summarized in Appendix A. The obtained recoverin forms were expressed and purified similarly to the rWT protein except that the cells were grown overnight at 30 °C and the mutants were eluted from the phenyl-Sepharose column with buffer A, containing 2–20 mM EDTA. The yield was 2–28 mg of protein per liter of cell culture depending on mutant.

### 3.3. Removal of Metal Ions from Recoverin

The contaminating Ca^2+^/Mg^2+^ ions were removed from recoverin samples using the gel-filtration method described by Blum et al. using Sephadex G-25 column equilibrated with a buffer [33]. An excess of Ca^2+^ chelator (10 mM EDTA) was added to recoverin before the gel-filtration. To protect the reduced recoverin from thiol oxidation, 5 mM of freshly prepared DTT was also added prior to the chromatography.

### 3.4. Chemical Crosslinking of Recoverin and Its Mutants

Protein (1.0 mg/mL) crosslinking with 0.02% glutaric aldehyde was performed in 20 mM Tricine-KOH pH 7.4, 50 mM KCl, 1 mM DTT; 1 mM EDTA (for apo-proteins), 1 mM MgCl_2_, 1 mM EGTA (for Mg^2+^-loaded proteins) or 1 mM CaCl_2_ (for Ca^2+^-loaded proteins). The reaction proceeded for 16 h at 20 °C and was stopped by addition of 4-fold volume of the buffer used in SDS polyacrylamide gel electrophoresis. The samples were subjected to SDS-PAGE (5% concentrating and 15% resolving gels; 5 µg of protein per lane) and stained with Coomassie Brilliant Blue R-250. The gels were scanned using Molecular Imager PharosFX Plus System (Bio-Rad Laboratories, Inc. (Hercules, VT, USA)) and analyzed by Quantity One software.

### 3.5. Circular Dichroism Measurements

Circular dichroism (CD) studies were carried out with a J-810 spectropolarimeter (JASCO, Inc.), equipped with a Peltier-controlled cell holder. The measurements were carried out at 15 °C. The instrument was calibrated with an aqueous solution of d-10-camphorsulfonic acid according to the manufacturer’s instruction. The cell compartment was purged with nitrogen (dew-point of –40 °C). The quartz cell with path-length of 1.00 mm was used for far-UV region measurements. Protein concentration was 3.3–3.7 µM. Buffer conditions: 10 mM Tricine-KOH, 50 mM KCl, 20 µM DTT pH 7.4; 1 mM EDTA (for apo-proteins), 1 mM MgCl_2_, 1 mM EGTA (for Mg^2+^-loaded proteins) or 1 mM CaCl_2_ (for Ca^2+^-loaded proteins). A small contribution of buffer was subtracted from experimental spectra. Band width was 2 nm, averaging time 2 s, and accumulation 3. The far-UV CD spectra were analyzed in 200–240 nm range using CDPro software package [34] (http://lamar.colostate.edu/~sreeram/CDPro/main.html). SELCON3, CDSSTR and CONTIN algorithms were also used for evaluation of the secondary structure fractions. SDP48 and SMP56 reference protein sets were used for these evaluations. The final secondary structure fractions represent averaged values.

Thermal denaturation of apo-forms of recombinant wild type human recoverin and its cluster mutants was monitored by far UV circular dichroism. Buffer conditions: 10 mM Tricine-KOH, 50 mM KCl, 1 mM EDTA, 20 µM DTT. Far-UV ellipticity θ at several wavelength (208, 216, and 222 nm) were measured in the temperature region from 15 °C to 98 °C in steps 2 °C (averaging time of the data at each temperature was 8 s). The temperature in the cuvette was equilibrated for 1.5 min. The measured ellipticity was converted into molar units according to the equation:[θ] = θ/(*C* × *l* × *z*)
where *C* is molar protein concentration; *l* is optical path length in mm; *z* is the number of amino acid residues in the protein (199 amino acids).

Mid-temperatures of thermal transitions in the proteins were evaluated by means of approximation of experimental data by sigmoidal function using OriginPro 9.0 (OriginLab Corporation (Northampton, MA, USA)).

### 3.6. Fluorescence Measurements

Fluorescence studies were performed with a Cary Eclipse spectrofluorometer (Varian, Inc. (Palo Alto, CA, USA)), equipped with a Peltier-controlled cell holder. Quartz cells with path-length of 10 mm were used. Protein concentrations were 3–12 µM. Fluorescence of Trp residues of recoverin was excited at 295 nm. In titration experiments all spectra were corrected for dilution according to the equation:F_λ_ × (1 − 10^−D^_0_)/(1 − 10^−D^_0_^/k^).
where F_λ_ denotes value of protein fluorescence at a wavelength λ, D_0_ is its absorption at the excitation wavelength, k is dilution coefficient.

### 3.7. Evaluation of Calcium Binding Parameters by Means of The Calcium Buffer Method

Calcium affinities of rWT human recoverin and its cluster mutants at 20 °C were determined using intrinsic protein fluorescence and calcium buffers. Protein concentration was 0.9–3.1. Buffer conditions: 50 mM MES-KOH pH 6.3 or HEPES-KOH pH 7.5, or MOPS-KOH pH 6.7–7.2, or Tricine-KOH pH 8.2, 1–2 mM DTPA-KOH, 1 mM DTT, 50 mM KCl; 20 μM–10 mM CaCl_2_. Previously calculated amounts of the standard CaCl_2_ solution were added to protein sample containing known concentration of EDTA to reach needed free calcium concentration [Ca^2+^]. The volumes of additions were calculated using the IBFluo v.1.1 program (IBI RAS, Pushchino, Russia). The effective Ca^2+^/Mg^2+^ binding constants of EDTA were calculated according to [35] using thermodynamic constants taken from IUPAC Stability Constants Database, SC-Database v.4.79, with corrections on temperature and ionic strength according to the Vant Hoff and Davis equations, respectively.

The dependence of protein fluorescence intensity, F_protein_, on free Ca^2+^ concentration was approximated by the sequential binding scheme suggested earlier for bovine recoverin [14] and various parvalbumins [36]. The theoretical curve was computed according to the following equations:F_protein_ = (*F_1_* + *F_2_* × f_1_) × α + *F_3_* × (1 – α ×(1+ f_1_),α = 1/(1 + f_1_ + f_1_ × f_2_),f_1_ = 10^(*pK1* - pCa)^,f_2_ = 10^(*pK2* - pCa)^.

The fit of the theoretical curve to experimental data was achieved by variation of *pK_1_*, *pK_2_* (logarithms of calcium dissociation constants) and *F_1_*, *F_2_, F_3_* (fluorescence intensities of the protein with zero, one, and two bound calcium ions, respectively) using Microcal OriginPro 8.0 (OriginLab Corporation (Northampton, MA, USA)). pCa is the logarithm of free calcium concentration. The accuracy of the binding constants evaluation did not exceed ± 1/4 orders of magnitude.

Protein fluorescence intensity, F_protein_, on free calcium concentration was approximated also by a cooperative binding model [37] using the following equations:F_protein_ = (*F_1_* + f × *F_2_*)/(1 + f),f = 10[(*logK* − pM) × *n*].

Theoretical curves were fitted to experimental data by variation of logarithm of equilibrium dissociation constant *logK*, *n* (binding cooperativity, Hill coefficient), *F_1_* and *F_2_* (fluorescence intensity of apo- and metal-loaded protein states, respectively) using Microcal OriginPro 9.0 (OriginLab Corporation) software. The accuracy of the binding constant evaluation did not exceed ± 1/4 orders of magnitude.

### 3.8. Determination of Metal Affinity of Recoverin from Protein Titration by Ca^2+^

Ca^2+^ affinity of some recoverin cluster mutants (F70A and Q46A) at 20 °C was estimated mainly as described in [37] from direct spectrofluorometric titration of the metal-depleted protein with CaCl_2_ standard solution. Recoverin concentration was 12–21 μM. Buffer conditions: 20 mM Tricine-KOH, 50 mM KCl, 1 mM DTT, pH 7.4. Intrinsic recoverin fluorescence intensities were corrected for the dilution effect via division of the experimental values by a factor of (1–10^−*Dexc*^), where *D_exc_* is the protein absorption at the excitation wavelength [25]. The experimental data were described by the sequential metal binding scheme:*K_a1_*P + M ↔ P∙M,*K_a2_*P∙M + M ↔ P∙M_2_,(I)
where P and M denote protein and metal ion, respectively; *K_a_**_1_* and *K_a2_* are equilibrium metal association constants for the two active EF-hands of recoverin.

The experimental calcium dependence of fluorescence intensity at 334 nm was globally fitted according to the schemes [I] using FluoTitr v.1.4 software (IBI RAS, Pushchino, Russia). Log *K_a1_*, log *K_a_**_2_* and fluorescence intensities of pure recoverin states P, PM and PM_2_ were used as fitting parameters. In the case when the experimental data were not described by *sequential metal binding* scheme (Q46A), a simple one-site binding scheme was used:*K*Protein + M ↔ Protein∙M,(II)
where M denotes metal ion, K is apparent equilibrium metal-binding constant.

### 3.9. The Equilibrium Centrifugation Assay of Recoverin Binding to Membranes

Bovine rod outer segments (ROS) and urea-washed photoreceptor membranes were prepared from frozen retinae according to the previously published method [38]. The binding of recoverin forms to photoreceptor membranes was studied using equilibrium centrifugation assay [39], with modifications described in [40]. Samples of rWT human recoverin or its mutants (30 μM) were mixed with bleached urea-washed photoreceptor membranes containing 10 μM rhodopsin in 20 mM Tris-HCl (pH 8.0), 150 mM NaCl, 20 mM MgCl_2_ and 1 mM DTT, with addition of either 250 µM CaCl_2_ or 2 mM EGTA in total volume of 50 μL. The probes were incubated at 37 ^°^C for 20 min and centrifuged at 14,000 rpm for 15 min. The supernatants were discarded and the pellets were dissolved in 50 μL of the sample buffer (125 mM Tris-HCl, pH 6.8, 4% (w/v) SDS, 20% (v/v) glycerol, 10% (v/v) β-mercaptoethanol, 0.004% (w/v) bromphenol blue) and analyzed by SDS-PAGE. The amounts of recoverin forms bound to photoreceptor membranes were evaluated by densitometric scanning of the corresponding bands in polyacrylamide gel.

### 3.10. Rhodopsin Phosphorylation Assay

GRK1 (rhodopsin kinase) was extracted from ROS as described elsewhere [41] or purchased from ThermoFisher Scientific (Waltham, MA, USA). GRK1 assay was performed as described in [42,43]. Briefly, 40 μM of rWT human recoverin or its mutants were mixed with 10 μM rhodopsin (urea-washed photoreceptor membranes) and 0.3–0.5 units of GRK1 in 20 mM Tris-HCl (pH 8.0), 100 mM NaCl, 1 mM [γ-^32^P]ATP, 1 mM DTT, 3 mM MgCl_2_, with addition of 250 mM CaCl_2_ or 2 mM EGTA. The reaction (25 min) was initiated by light illumination and terminated by addition of sample buffer for SDS-PAGE. The proteins were separated by polyacrylamide gel electrophoresis and ^32^P emission was registered by phosphorimaging radioautography.

### 3.11. Computational Evaluation of the Intrinsic Disorder Predisposition of Human Recoverin and Its Cluster Mutants

The intrinsic disorder propensity of human recoverin (UniProt ID: P35243) was evaluated by several commonly used disorder predictors, such as PONDR^®^ VSL2 [44] (which is one of the more accurate stand-alone disorder predictors [44,45,46]), PONDR^®^ VL3 [47] (which is a tool for accurate evaluation of long intrinsically disordered regions), PONDR^®^ VLXT [48] (a computational tool sensitive to the local sequence peculiarities that can be used for identification of disorder-based binding sites), IUPred platform [49] (that allow reliable evaluation of long and short disordered regions), and a metapredictor PONDR^®^ FIT [50] (which is more accurate than each of its component predictors, PONDR^®^ VLXT [48], PONDR^®^ VSL2 [44], PONDR^®^ VL3 [47], FoldIndex [51], and IUPred [49]). Outputs of these six tools were averaged to produce the mean disorder profiles for human recovering. The resulting mean disorder propensity for this protein based on the averaging of disorder profiles of individual predictors represents a way to increase the accuracy of predictive performance [46,50,52,53,54,55]. In these analyses, residues and regions are considered disordered or flexible if their predicted disorder scores are above 0.5 and between 0.2 and 0.5, respectively. We also utilized PONDR^®^ VSL2 to evaluate the effect of point cluster mutations (F35A, F83A, and Y86A in the N-terminal ‘black’ cluster, F106A, E169A, and F172A in the C-terminal ‘black’ cluster, F70A, Q46A, and F49A in the N-terminal ‘gray’ cluster, and W156A, K119A, and V122A in the C-terminal ‘gray’ cluster) on the intrinsic disorder predisposition of human recoverin. The mutation-induced changes in disorder propensity was further visualized by presenting the “difference disorder spectra” calculated by the subtracting the mean disorder profile of the wild type human recoverin from the disorder profiles of each of it cluster mutants.

## 4. Conclusions

The ‘black’ and ‘gray’ clusters were found in all members of the EF-hand calcium binding protein family [7]. Therefore, it was reasonable to suggest that these clusters should play some important structural and/or functional roles. One of the simplest ways to check this hypothesis is to use alanine screening; i.e., the analysis of the effects of substitutions of the amino acids of the clusters to alanines on the physical and functional properties of some representatives of the EF-hand protein family. We have started this study with the simplest EF-hand proteins, S100P protein and parvalbumin [8,9]. We have found that in both these proteins, amino acids of the ‘black’ cluster provide more essential contribution to the maintenance of structural and functional properties of the protein in comparison with the residues of the ‘gray’ cluster. The study of recoverin mutants was the next step in this series of experiments on elucidating the effects of the substitutions of the amino acids of the clusters to alanines on the physical and functional properties of the EF-hand protein. Recoverin has more complicated three-dimensional structure, where the four clusters are located close to each other, therefore the results obtained were not so evident as in the cases of S100P protein and parvalbumin. The mutations in the both ‘black’ and ‘gray’ clusters of recoverin changed the physical properties of the protein. At the same time, some mutations in the C-terminal part of recoverin had a much stronger effect on the structural properties of the protein compared to the mutations in the N-terminal part. The most of the N-terminal clusters mutations (except for F49A) suppressed membrane binding of Ca^2+^-loaded recoverin and hence its inhibitory activity towards GRK1. The negative effect of the substitutions of F35, Y86, and F49 (N-terminal part) on inhibitory activity of recoverin may be attributed to the participation of these residues in the hydrophobic pocket responsible for GRK1 binding. In contrast to the N-terminal mutations, the substitutions in C-terminal clusters of recoverin generally did not affect membrane binding of this protein, and the majority of the mutants (with the exception to F172A and W156A) exhibited effective GRK1 inhibition. The W156A mutant aberrantly activated rhodopsin phosphorylation regardless of the presence of calcium. These results show that the ‘black’ and ‘gray’ clusters of recoverin are very important for functioning of this protein.

## Figures and Tables

**Figure 1 molecules-24-02494-f001:**
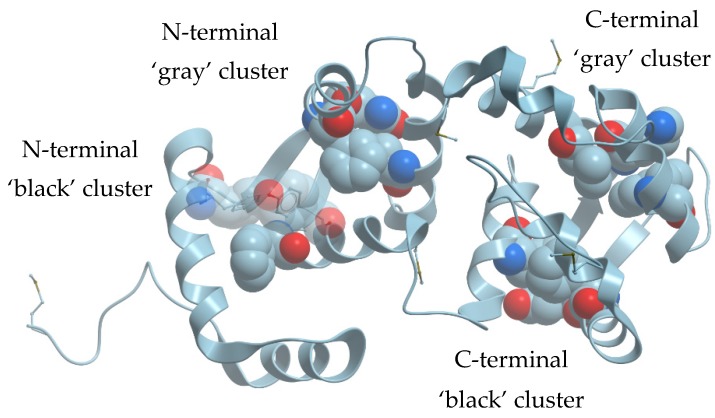
‘Black’ and ‘gray’ clusters in the N- and C-terminal parts of human recoverin (Protein Data Bank (PDB) ID: 2D8N).

**Figure 2 molecules-24-02494-f002:**
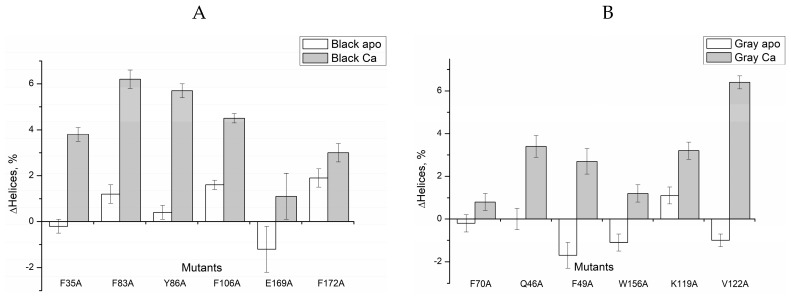
Effects of alanine substitutions in the ‘black’ (**A**) and ‘gray’ (**B**) clusters on α-helices content of recombinant wild type human recoverin in the absence of bound metal ions and in the presence of 1 mM CaCl_2_. Vertical bars show accuracy of α-helices content determination.

**Figure 3 molecules-24-02494-f003:**
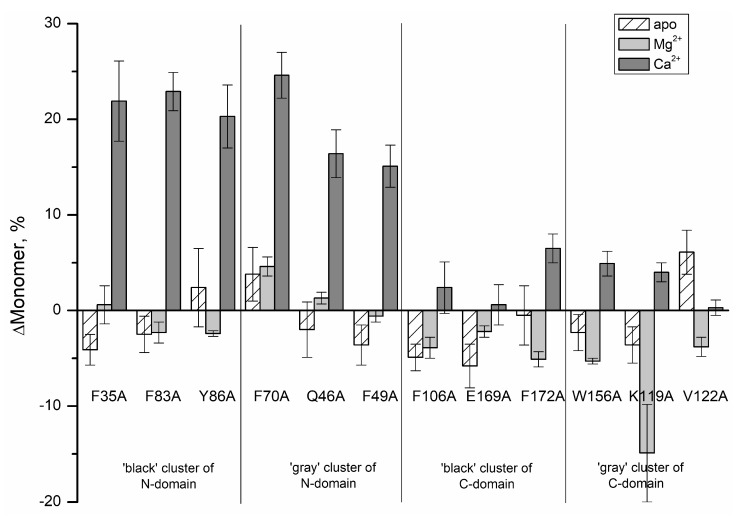
Effects of the alanine cluster mutations on monomer content for various states of recombinant wild type recoverin.

**Figure 4 molecules-24-02494-f004:**
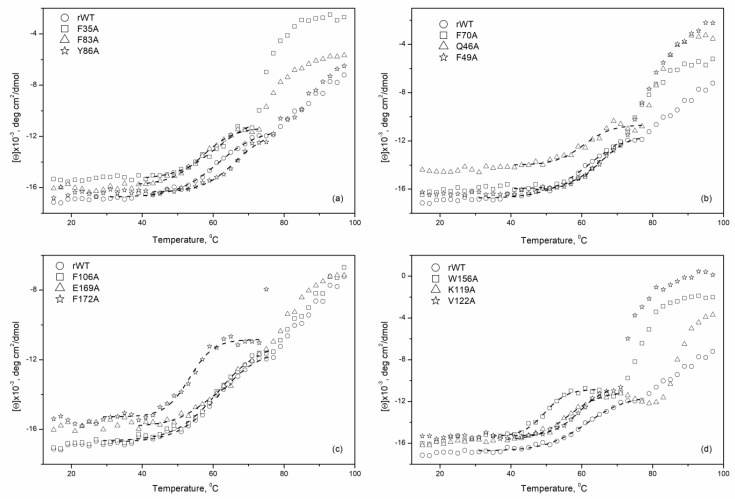
Temperature dependencies of molar ellipticity for apo-states of rWT human recoverin and its cluster mutants ((**a**)—‘black’ cluster of N-domain, (**b)**—‘gray’ cluster of N-domain, (**c)**—‘black’ cluster of C-domain, (**d**)—‘gray’ cluster of C-domain). 20 mM Tricin-KOH, pH 7.4, 50 mM KCl, 1 mM EDTA-KOH, 20 µM DTT. Protein concentration 3.3–3.7 μM. Theoretical curves computed according to a sigmoidal function and fitted to the experimental data are shown by dotted lines.

**Figure 5 molecules-24-02494-f005:**
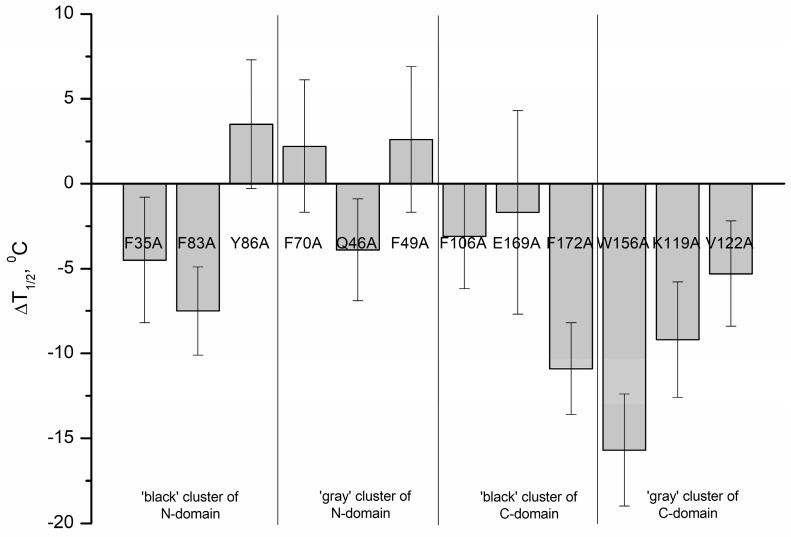
Effects of alanine substitutions in the ‘black’ and ‘gray’ clusters of rWT human recoverin on mid-temperature of its thermal transition in apo-state.

**Figure 6 molecules-24-02494-f006:**
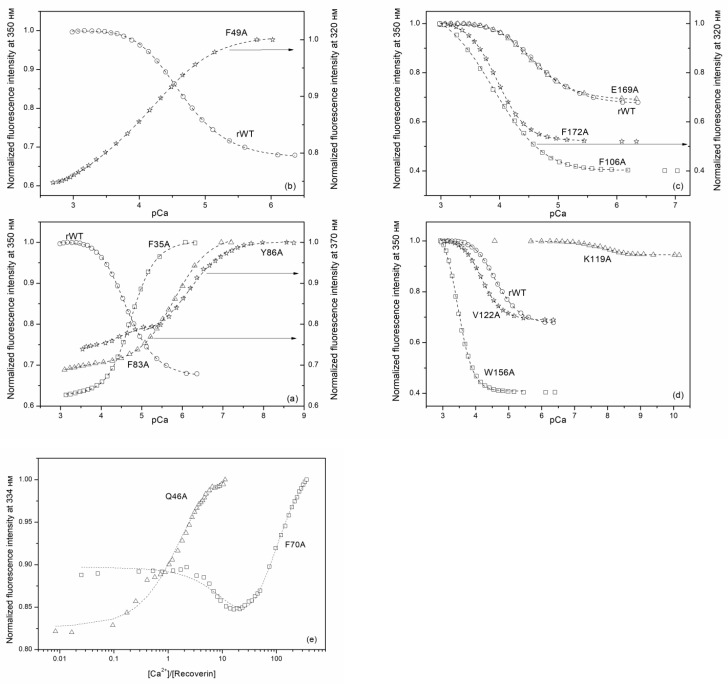
Spectrofluorometric Ca^2+^-titration of rWT human recoverin and its cluster mutants ((**a**)—‘black’ cluster of the N-domain, (**b**)—‘gray’ cluster of the N-domain, (**c**)—‘black’ cluster of the C-domain, (**d**)—‘gray’ cluster of the C-domain (the calcium buffer method), (**e**)—‘gray’ cluster of the N-domain (protein titration by Ca^2+^)) at 20 °C with the use of Ca^2+^-buffers. 50 mM 2-[N-morpholino] ethanesulfonic acid (MES)-KOH, pH 6.3 or 4-(2-hydroxyethyl)-1-piperazineethanesulfonic acid (HEPES)-KOH pH 7.5, or 3-(N-morpholino)propanesulfonic acid (MOPS)-KOH pH 6.7–7.2, or 20 mM Tricine-KOH pH 7.4–8.2, 50 mM KCl, 1–2 mM diethylenetriaminepentaacetic acid (DTPA)-KOH (in case of protein titration by Ca^2+^ was not used), 1 mM DTT. Protein concentration 0.9–3.1 µM, Ca^2+^ concentration 20 µM–10 mM. Dotted lines are computed according to the sequential or cooperative (only for F172A and W156A mutants) binding schemes and fitted to the experimental data.

**Figure 7 molecules-24-02494-f007:**
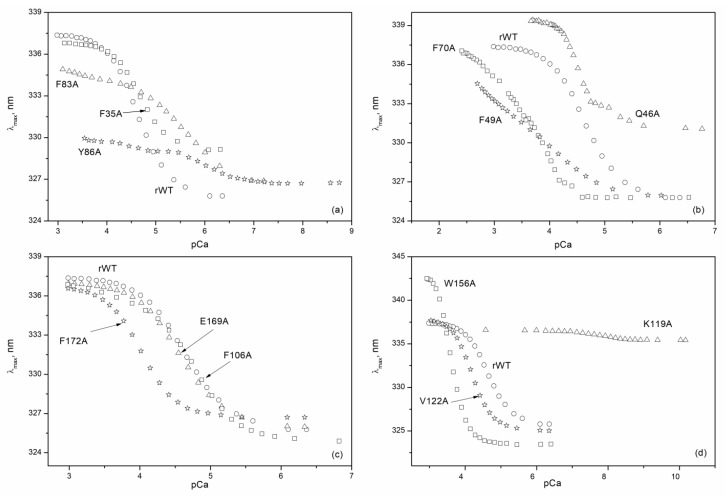
Ca^2+^-dependence of fluorescence spectrum maximum for rWT human recoverin and its cluster mutants ((**a**)—‘black’ cluster of the N-domain, (**b**)—‘gray’ cluster of the N-domain, (**c**)—‘black’ cluster of the C-domain, (**d**)—‘gray’ cluster of the C-domain) at 20 °C with the use of Ca^2+^-buffers.

**Figure 8 molecules-24-02494-f008:**
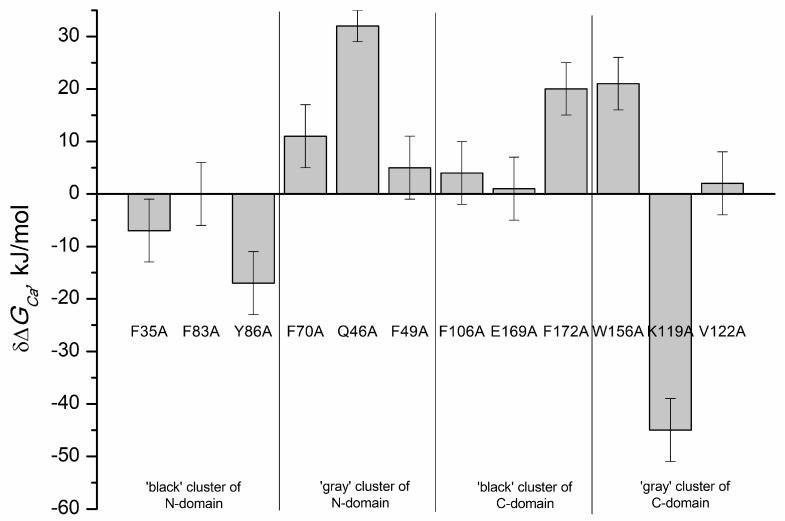
Effects of alanine substitutions in the ‘black’ and ‘gray’ clusters in rWT recoverin on the change of total free energy of protein induced by Ca^2+^ binding, *ΔG_Ca_* (−RT ln (*K_a,i_*•[H_2_O]) i = 1, 2) [26].

**Figure 9 molecules-24-02494-f009:**
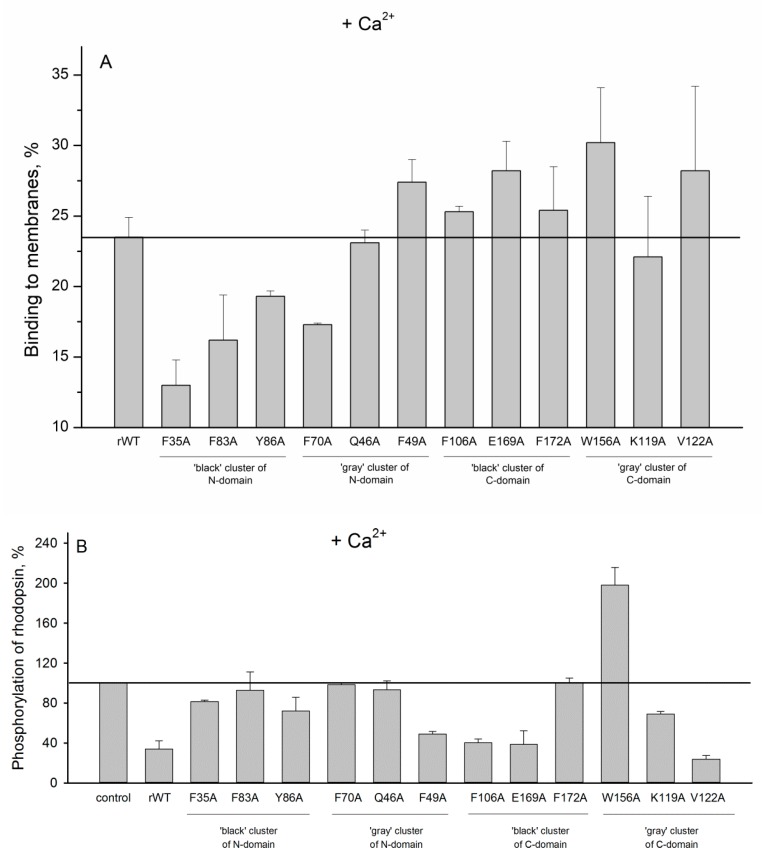
(**A**): The binding of calcium-loaded rWT human recoverin and its cluster mutants to membranes. (**B**): Effects of the alanine substitutions in the ‘black’ and ‘gray’ clusters in rWT human recoverin on phosphorylation of rhodopsin in the presence of calcium.

**Figure 10 molecules-24-02494-f010:**
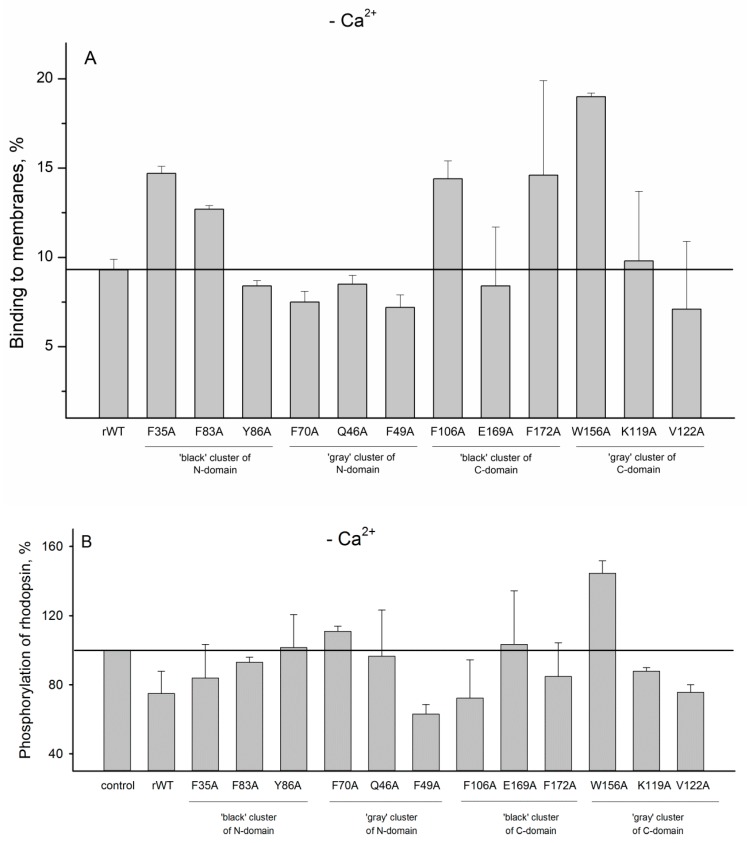
(**A**): The binding of metal-free rWT human recoverin and its cluster mutants to membranes. (**B**): Effects of the alanine substitutions in the ‘black’ and ‘gray’ clusters in rWT human recoverin on phosphorylation of rhodopsin in the absence of calcium ions.

**Figure 11 molecules-24-02494-f011:**
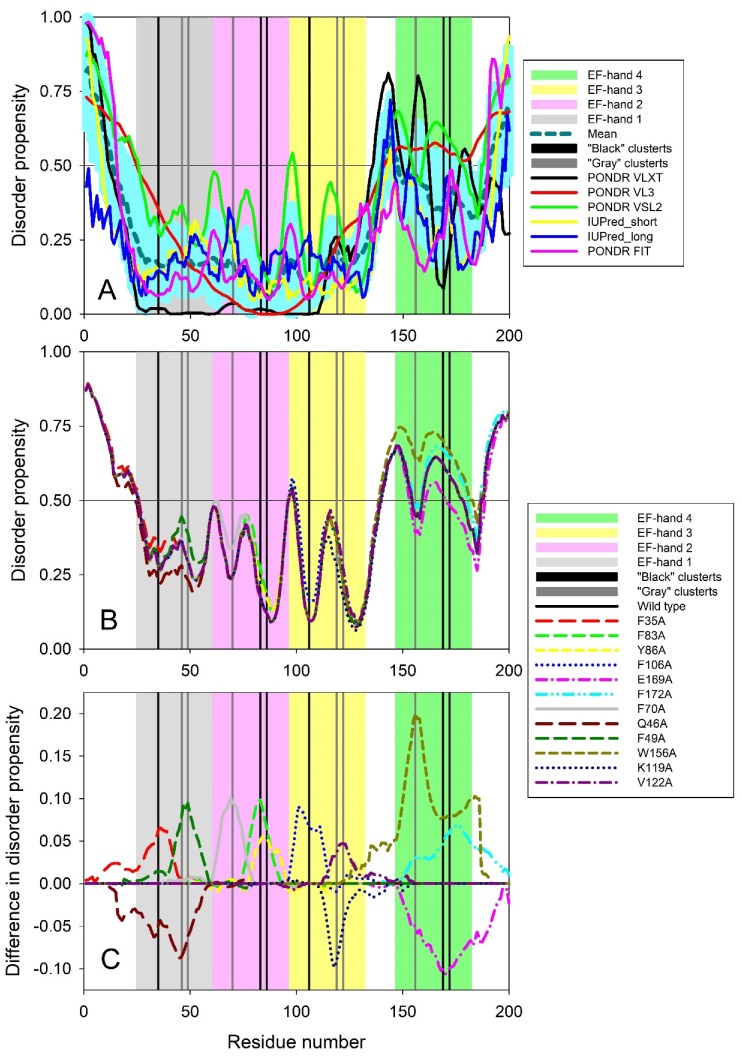
Intrinsic disorder predisposition of human recoverin (**A**) and effect of cluster mutations on intrinsic disorder propensity of this protein (**B**,**C**). In all plots, positions of four EF-hand motifs, residues 25–60, 61–96, 97–132, and 147–182 are shown by light gray, light pink, light yellow and light green bars, respectively. Positions of all mutations in ‘black’ and ‘gray’ clusters are shown by black and gray bars, respectively. Plot **A** shows results of a multiparametric analysis of the intrinsic disorder predisposition of human recoverin (UniProt ID: P35243) with several commonly used disorder predictors. *X*-axis shows the positions of the amino acids, whereas *Y*-axis represents the intrinsic disorder propensity of a given amino acid. Disorder profiles generated by PONDR^®^ VLXT, PONDR^®^ VL3, PONDR^®^ VSL2, PONDR^®^ FIT, and IUPred_short and IUPred_long are shown by black, red, green, pink, yellow, and blue lines, respectively. Thick dashed dark cyan line shows the mean disorder propensity calculated by averaging disorder profiles of individual predictors. Light cyan shadow around the mean disorder curve shows error distribution. In these analyses, the predicted intrinsic disorder scores above 0.5 are considered to correspond to the disordered residues/regions, whereas regions with the disorder scores between 0.2 and 0.5 are considered flexible. Plot **B** represents the PONDR^®^ VSL2-generated intrinsic disorder profiles of human recoverin protein and all its cluster mutants in a form of the per-residue mean disorder propensity calculated for each protein. Plot **C** shows “difference disorder spectra” calculated by subtracting PONDR^®^ VSL2-generated disorder profile of the wild type human recoverin from the mean disorder profiles of individual cluster mutants.

**Figure 12 molecules-24-02494-f012:**
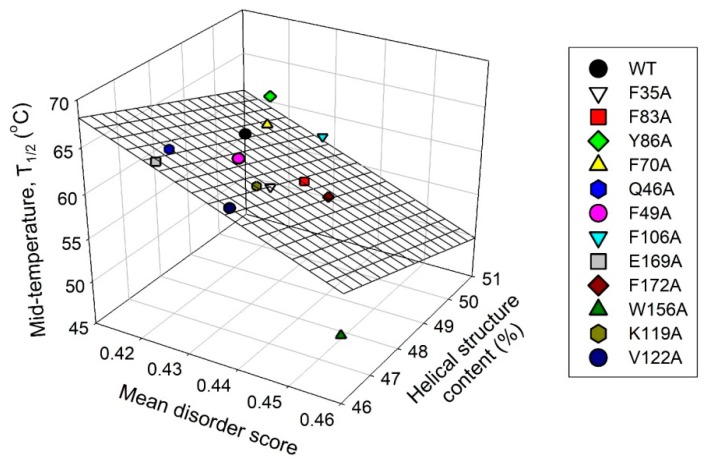
Correlation between the protein-average disorder propensity, helical propensity, and thermal stability of human recoverin and its cluster mutants.

**Table 1 molecules-24-02494-t001:** Secondary structure fractions estimated from the far-UV CD spectra of apo- (1 mM EDTA), Mg^2+^- (1mM MgCl_2_, 1 mM EGTA) and Ca^2+^-loaded (1 mM CaCl_2_) forms of recombinant wild type human recoverin and its cluster mutants.

Protein			Protein State	α-Helices %	β-Sheets %	Turns %	Unordered Structure
rWT			Apo-form	48.1 ± 0.1	8.9 ± 0.3	16.5 ± 0.2	26.8 ± 0.3
N-domain	‘black’ cluster	F35A		47.9 ± 0.2	8.6 ± 0.4	16.6 ± 0.2	26.7 ± 0.3
		F83A		49.3 ± 0.3	7.5 ± 0.3	15.9 ± 0.2	27.0 ± 0.1
		Y86A		48.5 ± 0.2	8.0 ± 0.2	15.7 ± 0.2	27.9 ± 0.3
	‘gray’ cluster	F70A		47.9 ± 0.3	8.4 ± 0.3	16.1 ± 0.3	27.6 ± 0.3
		Q46A		48.1 ± 0.4	7.9 ± 0.2	15.7 ± 0.2	27.8 ± 0.1
		F49A		46.4 ± 0.5	8.9 ± 0.3	16.4 ± 0.4	28.5 ± 0.5
C-domain	‘black’ cluster	F106A		49.7 ± 0.1	7.6 ± 0.4	15.8 ± 0.1	26.8 ± 0.4
		E169A		46.9 ± 0.9	8.7 ± 0.3	17.4 ± 0.2	27.3 ± 0.7
		F172A		50.0 ± 0.3	7.9 ± 0.2	16.4 ± 0.4	25.3 ± 0.3
	‘gray’ cluster	W156A		47.0 ± 0.3	9.0 ± 0.3	17.0 ± 0.1	27.4 ± 0.1
		K119A		49.2 ± 0.3	8.1 ± 0.3	16.1 ± 0.2	26.4 ± 0.2
		V122A		47.1 ± 0.2	8.9 ± 0.3	16.4 ± 0.1	27.9 ± 0.2
rWT			Mg^2+^	49.3 ± 0.1	7.9 ± 0.3	17.9 ± 0.2	25.3 ± 0.3
N-domain	‘black’ cluster	F35A		49.9 ± 0.2	8.2 ± 0.4	16.8 ± 0.2	25.4 ± 0.3
		F83A		51.1 ± 0.3	7.1 ± 0.3	15.9 ± 0.2	26.0 ± 0.1
		Y86A		50.9 ± 0.2	7.3 ± 0.2	16.6 ± 0.2	25.3 ± 0.3
N-domain	‘gray’ cluster	F70A		48.2 ± 0.3	8.0 ± 0.3	17.4 ± 0.3	26.6 ± 0.3
		Q46A		50.9 ± 0.4	7.8 ± 0.2	16.0 ± 0.2	25.3 ± 0.1
		F49A		48.4 ± 0.5	7.9 ± 0.3	16.8 ± 0.4	27.1 ± 0.5
C-domain	‘black’ cluster	F106A		53.2 ± 0.1	7.0 ± 0.4	16.2 ± 0.1	24.2 ± 0.4
		E169A		49.8 ± 0.9	7.4 ± 0.3	17.1 ± 0.2	26.2 ± 0.7
		F172A		51.1 ± 0.3	7.9 ± 0.2	15.7 ± 0.4	25.5 ± 0.3
	‘gray’ cluster	W156A		49.1 ± 0.3	7.4 ± 0.3	17.4 ± 0.1	26.3 ± 0.1
		K119A		51.3 ± 0.3	7.8 ± 0.3	15.5 ± 0.2	25.4 ± 0.2
		V122A		48.9 ± 0.2	7.9 ± 0.3	16.7 ± 0.1	26.5 ± 0.2
rWT			Ca^2+^	51.2 ± 0.1	7.0 ± 0.3	16.0 ± 0.2	25.7 ± 0.3
N-domain	‘black’ cluster	F35A		55.0 ± 0.2	7.5 ± 0.4	14.3 ± 0.2	23.5 ± 0.3
		F83A		57.4 ± 0.3	5.4 ± 0.3	14.4 ± 0.2	22.5 ± 0.1
		Y86A		56.9 ± 0.2	5.4 ± 0.2	13.9 ± 0.2	23.5 ± 0.3
	‘gray’ cluster	F70A		52.0 ± 0.3	7.0 ± 0.3	15.8 ± 0.3	25.1 ± 0.3
		Q46A		54.6 ± 0.4	7.0 ± 0.2	15.4 ± 0.2	23.0 ± 0.1
		F49A		53.9 ± 0.5	6.9 ± 0.3	15.9 ± 0,4	22.8 ± 0.5
C-domain	‘black’ cluster	F106A		55.7 ± 0.1	5.8 ± 0.4	14.0 ± 0.1	24.2 ± 0.4
		E169A		52.3 ± 0.9	6.8 ± 0.3	15.3 ± 0.2	25.3 ± 0.7
		F172A		54.2 ± 0.3	6.6 ± 0.2	15.5 ± 0.4	23.4 ± 0.3
	‘gray’ cluster	W156A		52.4 ± 0.3	7.1 ± 0.3	15.5 ± 0.1	24.9 ± 0.1
		K119A		54.4 ± 0.3	6.8 ± 0.3	15.4 ± 0.2	23.5 ± 0.2
		V122A		57.6 ± 0.2	6.0 ± 0.3	14.4 ± 0.1	22.0 ± 0.2

**Table 2 molecules-24-02494-t002:** Distribution of oligomeric forms of recombinant wild type human recoverin and its cluster mutants obtained from experiments on chemical crosslinking by glutaric aldehyde (0.02%) at 20 °C. Protein concentration 1 mg/mL. 20 mM Tricine-KOH pH 7.4, 50 mM KCl, 1 mM dithiothreitol (DTT); 1 mM EDTA (for apo-proteins), 1mM MgCl_2_, 1 mM EGTA (for Mg^2+^-loaded proteins) or 1 mM CaCl_2_ (for Ca^2+^-loaded proteins).

Protein			Protein State	Monomer 20 kDa, %	Dimer 50 kDa, %	Trimer 70–100 kDa, %	Multimer 100–250 kDa, %
rWT			Apo-state	66.9 ± 1.1	33.1 ± 1.1	-	-
N-domain	‘black’ cluster	F35A		62.8 ± 0.5	37.2 ± 0.5	-	-
		F83A		64.4 ± 0.8	35.6 ± 0.8	-	-
		Y86A		69.3 ± 3.0	30.7 ± 3.0	-	-
	‘gray’ cluster	F70A		70.7 ± 1.7	29.3 ± 1.7	-	-
		Q46A		64.9 ± 1.8	35.1 ± 1.8	-	-
		F49A		63.3 ± 1.0	36.7 ± 1.0	-	-
C-domain	‘black’ cluster	F106A		62.0 ± 0.3	38.0 ± 0.3	-	-
		E169A		61.1 ± 1.2	38.9 ± 1.2	-	-
		F172A		66.4 ± 2.0	33.6 ± 2.0	-	-
	‘gray’ cluster	W156A		64.6 ± 0.8	35.4 ± 0.8	-	-
		K119A		63.3 ± 0.8	36.7 ± 0.8	-	-
		V122A		73.0 ± 1.2	27.0 ± 1.2	-	-
rWT			Mg^2+^-loaded	68.45 ± 0.09	31.55 ± 0.09	-	-
N-domain	‘black’ cluster	F35A		69.1 ± 1.9	30.9 ± 1.9	-	
		F83A		66.2 ± 1.0	33.8 ± 1.0	-	
		Y86A		66.1 ± 0.2	33.9 ± 0.2	-	
	‘gray’ cluster	F70A		73.1 ± 0.9	26.9 ± 0.9	-	-
		Q46A		69.8 ± 0.5	30.2 ± 0.5	-	-
		F49A		67.9 ± 0.5	32.1 ± 0.5	-	-
C-domain	‘black’ cluster	F106A		64.6 ± 1.0	35.4 ± 1.0	-	-
		E169A		66.3 ± 0.5	33.7 ± 0.5	-	-
		F172A		63.4 ± 0.7	36.6 ± 0.7	-	-
	‘gray’ cluster	W156A		63.2 ± 0.2	36.8 ± 0.2	-	-
		K119A		53.6 ± 5.0	46.4 ± 5.0	-	-
		V122A		64.7 ± 0.9	35.3 ± 0.9	-	-
rWT			Ca^2+^-loaded	15.5 ± 0.6	30.9 ± 0.4	32.0 ± 1.1	21.55 ± 0.09
N- domain	‘black’ cluster	F35A		37.4 ± 3.6	31.7 ± 0.3	17.78 ± 0.07	13.2 ± 3.3
		F83A		38.4 ± 1.4	28.7 ± 0.8	19.7 ± 0.4	13.2 ± 0.3
		Y86A		35.8 ± 2.7	28.5 ± 1.4	21.5 ± 0.7	14.2 ± 0.6
	‘gray’ cluster	F70A		40.1 ± 1.8	29.3 ± 1.1	19.1 ± 0.5	11.4 ± 0.2
		Q46A		31.9 ± 1.9	27.3 ± 1.5	22.3 ± 1.0	18.4 ± 0.6
		F49A		30.6 ± 1.6	25.5 ± 0.5	22.8 ± 1.3	21.08 ± 0.09
C-domain	‘black’ cluster	F106A		17.9 ± 2.1	27.11 ± 0.08	30.0 ± 1.4	25.0 ± 0.8
		E169A		16.1 ± 1.5	27.0 ± 0.3	29.6 ± 0.3	27.2 ± 0.9
		F172A		22.0 ± 0.9	29.3 ± 0.2	28.8 ± 0.8	20.0 ± 0.3
	‘gray’ cluster	W156A		20.4 ± 0.7	25.0 ± 1.1	24.8 ± 1.0	29.8 ± 0.9
		K119A		19.5 ± 0.4	24.4 ± 0.6	24.7 ± 0.2	31.38 ± 0.05
		V122A		15.8 ± 0.2	27.1 ± 1.3	30.3 ± 0.6	26.7 ± 2.1

**Table 3 molecules-24-02494-t003:** Mid-temperatures, T_1/2_, of the thermal transitions of apo-states of rWT human recoverin and its cluster mutants monitored by far-UV circular dichroism method. The experimental data for ellipticity at fixed wavelengths (208, 216, and 222 nm) were approximated by sigmoidal curves and T_1/2_ is a mean value for these three curves.

		Protein	T_1/2_, °C			Protein	T_1/2_, °C
N-terminal domain	‘black cluster’	rWT	64.3 ± 2.5	C-terminal domain	‘black cluster’		
		F35A	59.8 ± 1.2			F106A	61.2 ± 0.6
		F83A	56.8 ± 0.1			E169A	62.6 ± 3.5
		Y86A	67.8 ± 1.3			F172A	53.4 ± 0.2
	‘gray cluster’	F70A	66.5 ± 1.4		‘gray cluster’	W156A	48.6 ± 0.8
		Q46A	60.4 ± 0.5			K119A	55.1 ± 0.9
		F49A	66.9 ± 1.8			V122A	59.0 ± 0.6

**Table 4 molecules-24-02494-t004:** Logarithms of equilibrium calcium binding constants of rWT human recoverin and its cluster mutants at 20 °C, evaluated according to the sequential and cooperative binding schemes using intrinsic fluorescence, calcium buffers method and protein titration by Ca^2+^.

Protein		Sequential Binding Model		Cooperative Binding Model	
		log K_1_	log K_2_	log K	n
rWT		4.5	4.1	4.6	1.3
N-terminal domain					
‘black’ cluster	F35A	4.6	4.8	4.7	1.5
	F83A	5.8	2.8	5,8	0.9
	Y86A	6.2	4.5	n/d	n/d
‘gray’ cluster	F70A	3.2	3.4	n/a	n/a
	Q46A	4.7		n/a	n/a
	F49A	4.4	3.3	4.1	0.8
C-terminal domain					
‘black’ cluster	F106A	3.9	4.0	3.9	1.4
	E169A	4.4	4.1	4.6	1.4
	F172A	n/d	n/d	4.0	1.7
‘gray’ cluster	W156A	n/d	n/d	3.5	1.9
	K119A	8.7	7.9	7.9	1.1
	V122A	4.0	4.2	4.2	1.4

n/d, not determined because of bad fit; n/a, not available.

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
