# Peer review of "Experimental Insight into the Structural and Functional Roles of the ‘Black’ and ‘Gray’ Clusters in Recoverin, a Calcium Binding Protein with Four EF-Hand Motifs"

_molecules, 2019, doi:10.3390/molecules24132494_

Round 1
Reviewer 1 Report
The manuscript provides a comprehensive analysis of a large number of mutations in the Ca2+-binding protein, recoverin to help understand the energetics of Ca2+-induced conformational changes. The experimental data and analysis all appear to be satisfactory. However, the biological significance and importance of this study seems unclear. What important questions were answered by the study? What is the functional significance of the "black" and "grey" clusters? And how do these clusters connect to the biological function of the protein? There needs to be more discussion to better explain the significance of these clusters and better explain the rationale of the study.
Author Response
Comments and Suggestions for Authors (Reviewer 1):
The manuscript provides a comprehensive analysis of a large number of mutations in the Ca2+-binding protein, recoverin to help understand the energetics of Ca2+-induced conformational changes. The experimental data and analysis all appear to be satisfactory.
REPLY: We are thankful to this reviewer for careful reading of the manuscript and for useful suggestions.
Some considerations on the functional significance of the "black" and "grey" clusters for recoverin are presented in the last paragraph on the Page 21-22.
However, the biological significance and importance of this study seems unclear. What important questions were answered by the study?
Reply: The ‘black’ and ‘gray’ clusters were found in all members of the EF-hand calcium binding protein family [7]. Therefore, it was reasonable to suggest that these clusters should play some important structural and/or functional roles. One of the simplest ways to check this hypothesis is to use alanine screening; i.e., the analysis of the effects of substitutions of the amino acids of the clusters to alanines on the physical and functional properties of some representatives of the EF-hand protein family. We have started this study with the simplest EF-hand proteins, S100P protein and parvalbumin [8, 9]. We have found that in both these proteins, amino acids of the ‘black’ cluster provide more essential contribution to the maintenance of structural and functional properties of the protein in comparison with the residues of the ‘gray’ cluster. The study of recoverin mutants was the next step in this series of experiments on elucidating the effects of the substitutions of the amino acids of the clusters to alanines on the physical and functional properties of the EF-hand protein. Recoverin has more complicated three-dimensional structure, where the four clusters are located close to each other, therefore the results obtained were not so evident as in the cases of S100P protein and parvalbumin. Nevertheless, we found that all the clusters are important for structure and function of recoverin.
What is the functional significance of the "black" and "grey" clusters?
Reply: In our previous works we have found that in the EF-hand proteins with simplier structure, parvalbumin and S100P protein, amino acids of the 'black' cluster in provide more essential contribution to the maintenance of structural and functional properties of these proteins in comparison with the residues of the 'gray' cluster [8,9]. It is not the case in recoverin.The mutations in the ‘black’ and ‘gray’ clusters of recoverin both changed the physical properties of the protein. At the same time, some mutations in the C-terminal part of recoverin had a much stronger effect on the structural properties of the protein compared to the mutations in the N-terminal part.
And how do these clusters connect to the biological function of the protein? There needs to be more discussion to better explain the significance of these clusters and better explain the rationale of the study.
REPLY: The most of the N-terminal clusters mutations (except for F49A) suppressed membrane binding of Ca2+-loaded recoverin and hence its inhibitory activity towards GRK1. The negative effect of the substitutions of F35, Y86, and F49 (N-terminal part) on inhibitory activity of recoverin may be attributed to the participation of these residues in the hydrophobic pocket responsible for GRK1 binding. In contrast to the N-terminal mutations, the substitutions in C-terminal clusters of recoverin generally did not affect membrane binding of this protein, and the majority of the mutants (with the exception to F172A and W156A) exhibited effective GRK1 inhibition. The W156A mutant aberrantly activated rhodopsin phosphorylation regardless of the presence of calcium. These results show that the ‘black’ and ‘gray’ clusters of recoverin are very important for functioning of this protein.
Reviewer 2 Report
This manuscript describes the structural roles of two conserved clusters in recoverin. The clusters are observed commonly in EF-hands.
Structural and functional analysis of Ala substituted mutant has been carried out comprehensively in the clusters.
minor points
CD spectra might be affected by the mutation of aromatic residues. Figure 2 and 3 should be combined to show the calcium dependent change of each mutant.
Some typos including Russian characters should be corrected. For example, Page 23, equation next to line 555. The legend of Figure 7, 31 MK M at line 213.
Author Response
This manuscript describes the structural roles of two conserved clusters in recoverin. The clusters are observed commonly in EF-hands. Structural and functional analysis of Ala substituted mutant has been carried out comprehensively in the clusters.
REPLY: We are thankful to this reviewer for careful reading of the manuscript and for useful suggestions.
Minor points:
1) CD spectra might be affected by the mutation of aromatic residues. Figure 2 and 3 should be combined to show the calcium dependent change of each mutant.
REPLY: We have combined Figures 2 and 3 on Page 4 and changed the numeration of the rest figures.
2) Some typos including Russian characters should be corrected.
REPLY: Russian characters were changed (Page 25, equation next to line 588. The legend of Figure 6, 3.1 мкM at line 211).
Besides, the following changes were done:
new e-mail address on Page 1, line 20;
‘multimerization’ instead ‘multimetization’ on Page 4 and 8, at lines 146 and 169, respectively;
‘increase’ instead of ‘decrease’ on Page 12, line 237;
space between words ‘black’ cluster in Table 4 on Page 13;
in the legend of Figure 11 ‘EF-hand 2’ instead of ‘EF-had 2’ on Page 18;
Figure 12 was added on Page 20;
‘and’ instead of ‘amd’ on Page 25, at line 607.